# Manufacturing and Characterization of Highly Environmentally Friendly Sandwich Composites from Polylactide Cores and Flax-Polylactide Faces

**DOI:** 10.3390/polym13030342

**Published:** 2021-01-21

**Authors:** Diego Lascano, Rene Guillen-Pineda, Luis Quiles-Carrillo, Juan Ivorra-Martínez, Rafael Balart, Nestor Montanes, Teodomiro Boronat

**Affiliations:** Technological Institute of Materials (ITM), Universitat Politècnica de València (UPV), Plaza Ferrándiz y Carbonell 1, 03801 Alcoy, Spain; reguipi@upv.es (R.G.-P.); luiquic1@epsa.upv.es (L.Q.-C.); rbalart@mcm.upv.es (R.B.); nesmonmu@upvnet.upv.es (N.M.); tboronat@dimm.upv.es (T.B.)

**Keywords:** PLA honeycomb core, eco-friendly sandwich structures, three-point bending test, hot compression molding

## Abstract

This work focuses on the manufacturing and characterization of highly environmentally friendly lightweight sandwich structures based on polylactide (PLA) honeycomb cores and PLA-flax fabric laminate skins or facings. PLA honeycombs were manufactured using PLA sheets with different thicknesses ranging from 50 to 500 μm. The PLA sheets were shaped into semi-hexagonal profiles by hot-compression molding. After this stage, the different semi-hexagonal sheets were bonded together to give hexagonal panels. The skins were manufactured by hot-compression molding by stacking two Biotex flax/PLA fabrics with 40 wt% PLA fibers. The combined use of temperature (200 °C), pressure, and time (2 min) allowed PLA fibers to melt, flow, and fully embed the flax fabrics, thus leading to thin composite laminates to be used as skins. Sandwich structures were finally obtained by bonding the PLA honeycomb core with the PLA-flax skins using an epoxy adhesive. A thin PLA nonwoven was previously attached to the external hexagonal PLA core, to promote mechanical interlock between the core and the skins. The influence of the honeycomb core thickness on the final flexural and compression properties was analyzed. The obtained results indicate that the core thickness has a great influence on the flexural properties, which increases with core thickness; nevertheless, as expected, the bonding between the PLA honeycomb core and the skins is critical. Excellent results have been obtained with 10 and 20 mm thickness honeycombs with a core shear of about 0.60 and facing bending stresses of 31–33 MPa, which can be considered as candidates for technical applications. The ultimate load to the sample weight ratio reached values of 141.5 N·g^−1^ for composites with 20 mm thick PLA honeycombs, which is comparable to other technical composite sandwich structures. The bonding between the core and the skins is critical as poor adhesion does not allow load transfer and, while the procedure showed in this research gives interesting results, new developments are necessary to obtain standard properties on sandwich structures.

## 1. Introduction

The demand for low-weight and high-rigidity materials in high-performance sectors has given way to the development of composite materials [1]. Among others, sandwich structures deserve special attention due to their use in a wide variety of sectors, which include conventional packaging with corrugated craft cores [2], and high-performance applications in aerospace [3,4], automotive [5], aeronautics [6], lightweight civil infrastructure [7], and so on. Sandwich panels are composed of a lightweight core and two (top and bottom) skins [8]. The most common cores in sandwich panels are processed woods (i.e., balsa wood), thermoplastic and thermosetting polymer foams, and honeycomb structures. Honeycombs can be obtained from a wide variety of materials such as metals (steel, aluminum, titanium) and thermoplastic polymers such as polypropylene or aramid. The skins or faces are usually made of lightweight and stiff materials such as aluminum or fiber-reinforced polymers (FRP) [9,10]. The most critical part in a sandwich structure is the skin–core interface, which plays a key role in load transfer. To enhance this interface, adhesives, fiber mats, or thin sheets are used [11]. The final composition of a sandwich structure depends on the target application. For example, polymer foam cores are generally used in car flooring, boat parts, as well as turbine blades, as they have good rigidity, high strength, and resistance to fatigue and temperature [12,13,14,15]. Feng and Aymerich [13] studied the effect of the density of a polyvinyl chloride (PVC) foam core on sandwich panels with carbon/epoxy-laminated facings, resulting in a composite structure with excellent stiffness. They demonstrated an increasing tendency in stiffness with increasing core density. One of the main drawbacks of sandwich-type composites is that they are expensive due to their manufacturing process. On the other hand, sandwich-type composites are difficult to handle with conventional manufacturing processes for composite materials such as vacuum-assisted resin infusion molding (VARIM) or resin transfer molding (RTM). In the last years, important advances have been carried out in the field of composite sandwich structures to overcome or palliate this. Lascano et al. [16] reported the manufacturing of a sandwich-type composite with basalt-flax hybrid laminates skins and a polypropylene-permeating nonwoven core (Lantor Soric© XF) with a hexagonal groove, which allows conventional manufacturing by the resin infusion processes. These new core materials combine lightness, processability, and stiffness.

Sandwich composite structures, particularly with honeycomb cores, are widely used because they offer excellent out-of-plane properties (i.e., bending, compression), compared to conventional composite laminates. This feature, together with their lightness because of the use of a very lightweight core material, have positioned sandwich structures as high-performance materials for applications in aerospace, automotives, and the sports and marine industry, among others [17]. Initially, sandwich structures with a honeycomb core were only used in aeronautics and aerospace industries due to their extremely positive stiffness-to-weight ratio (ailerons, flaps, rudders, and so on). Nevertheless, their production was laborious and, therefore, expensive, which made them unsuitable for mass production. Aluminum has been, with difference, the most common material for sandwich skins or cores in high-performance composites [9,18]. Nevertheless, with the development of engineering and high-performance polymers, a new series of honeycombs can be found as they also provide high stiffness with lower weight compared to aluminum. Among others, it is worth remarking the increasing use of polypropylene (PP) honeycombs for conventional applications, which provide good rigidity with a considerable decrease in weight, as PP is one of the lightest polymers [19]. For high-performance composites, aramid honeycombs are widely used due to an excellent balance between lightness, stiffness, cost, and temperature resistance [6].

Despite the fact that composite panels usually combine the optimum materials for the core, bonding, and skins, some composite panels are manufactured entirely from one material such as lightweight composite panels consisting of PP skins and PP honeycombs. These composite panels have become very popular in the packaging and construction industries [20,21]. The use of high-performance composite laminates with glass fiber (GF) and carbon fiber (CF) has widened the performance of sandwich structures for applications requiring high strength at low weight, because unlike metallic materials, they offer a better strength-to-weight ratio [22,23,24,25].

The increasing awareness of environmental issues has led to the search of new environmentally friendly materials. This situation is more pronounced in the polymers and composites industries, which are highly petroleum-dependent. In the last years, biobased materials are acquiring relevance as they can be obtained from natural resources. Despite there still being a long way to go with biobased polymers and composites, some materials have been positioned as excellent candidates for the replacement of their petrochemical counterparts [26,27]. Companies such as EconCore have developed a revolutionary technology, ‘ThermHex’ [28], which allows the serial production of honeycomb cores in a single sheet of thermoplastics material such as polypropylene (PP), polyethylene terephthalate (PET), and polycarbonate (PC). This process contributes to reduction in both the amount of material used, waste material, and energy consumption, all contributing to reducing the carbon footprint [29]. This technology has given way to the development of fully bio-based honeycomb materials by using polylactide (PLA), obtained from natural resources [1,25]. PLA is a thermoplastic material obtained from renewable sources through sugar fermentation or from starch-rich products such as potatoes or sugar cane. PLA offers interesting properties such as biodegradability (actually, disintegration in controlled compost soil), biocompatibility, good tensile strength, good stiffness, and shape memory [30]. In addition, it can be manufactured by conventional processing techniques such as injection molding, extrusion, hot compression molding, and 3D printing. For these reasons, the use of PLA has remarkably grown in the last decade, mainly in the packaging industry [31,32], and biomedicine [33,34]. The use of PLA can positively contribute to develop a new series of environmentally friendly composite materials or green composites. In fact, PLA is currently used as matrix in wood plastic composites with natural fibers and/or lignocellulosic fillers [35], and it is increasingly used in several components of composite sandwiches such as 3D-printed honeycombs, adhesion nonwovens, and skins on different composite panel configurations [36].

Furthermore, by replacing the currently used synthetic fibers with natural fibers, materials with balanced properties and environmental efficiency can be obtained. Among different natural fibers such as jute, hemp, sisal, and cotton, flax fibers are increasingly being used in green composites. Flax fibers offer superior properties to other natural fibers [37] and represent a good choice for green composites. Flax fibers are mainly composed of cellulose (60–85%) and hemicellulose (14–20%). The use of cellulosic materials as reinforcements provides low density, renewability, and/or biodegradability with a relatively low cost [38]. Flax fibers offer interesting mechanical properties, but there is very high heterogeneity on these properties depending on the flax variety, crop conditions, fiber diameter, position in the plant, and fiber length, among others [39]. The gauge length of the fiber plays a key role in the mechanical properties as described by Amroune et al. [40]. They reported a dramatic change in tensile strength from 1415 MPa for flax fibers with a gauge length of 10 mm down to values of 431 MPa for a gauge length of 500 mm, thus giving clear evidence of the influence of the gauge length on final performance. The same was observed for the tensile modulus, with changes from 54.20 to 31.45 GPa for gauge lengths of 10 and 500 mm, respectively.

Nevertheless, green composites with natural fibers offer an important drawback that is related to the hydrophilic nature of the reinforcement, which leads to an undesirable water uptake process that is responsible for dimensional changes and ageing [41]. This drawback can be minimized by using chemical or physical surface treatment on the fiber or by using coupling agents [42,43]. Despite flax and other bast fibers such as ramie, jute, kenaf, and hemp finding increasing applications as reinforcements in construction and building, the automotive sector and the sports industry, as reported by Sadrmanesh et al. [44], the use of flax and other bast fibers must face some challenges related to standardized mechanical properties for natural fibers (highly dependent on the crop conditions and manufacturing stages), water absorption, poor tensile and impact properties, and price fluctuations, compared to conventional glass, carbon, or aramid fibers [45].

Research conducted by Nickels [46] has shown the potential of PLA-derived materials for use in automobile parts, showing the potential of PLA honeycombs to obtain lightweight materials with balanced mechanical properties and, the most important, contributing to sustainable development.

This work aims to develop highly environmentally friendly sandwich structures for technical applications. To achieve this, this research explores the possibilities of PLA-based honeycomb cores with different thicknesses and the use of flax fabric-PLA laminates as sandwich skins or outer faces. The effect of the PLA honeycomb thickness on mechanical properties (flexural and compression) is evaluated.

## 2. Materials and Methods

### 2.1. Materials

A commercial aluminum block type 6082-T6 supplied by Broncesval SL (Valencia, Spain) was necessary for the manufacture of the mold to be used in the production of the honeycomb cores.

The semi-hexagonal-shape cell sheets were manufactured using a commercial polylactide (PLA) Ingeo^TM^ 6201D grade supplied by NatureWorks LLC (Minnetonka, MN, USA). This PLA grade has a glass transition temperature (T_g_) between 55 and 60 °C and a melting point located in the 155–170 °C range. Its melt flow index ranges between 15 and 30 g/10 min at 210 °C. A transparent ethyl cyanoacrylate adhesive, Loctite^®^ 401, supplied by Henkel (Düsseldorf, Germany) was used to bond the two semi-hexagonal-shaped sheets to obtain the honeycomb. This adhesive provides a rapid bonding and it is recommended to use on plastics. It has a Brookfield viscosity of 100–120 mPa s at 25 °C (screw 1; rotating speed = 30 rpm).

The outer sheets (sandwich skins) were manufactured with Biotex Flax/PLA. This fabric is a commingled textile composed of natural flax fiber and polylactide (PLA). It was supplied by SCABRO tooling and composites (Katwijk, Nederland) in the weave style of 2 × 2 twill and a surface density of 400 g·m^−2^. These commingled textiles can be converted into rigid engineering parts by applying a combination of pressure and temperature to allow PLA to flow and embed the flax reinforcement. The sandwich skins consisted of two stacked Biotex Flax/PLA fabrics. After being subjected to hot compression molding, the thin laminate offered a flexural modulus of 7.8 GPa and a flexural strength of 131 MPa.

A partially biobased epoxy resin based on soybean oil was used to bond the honeycomb and the skins. This resin was an Ecopoxy fast-curing system supplied by ECOEPOXY (Morris, MB, Canada) with an epoxy-to-hardener (amine-based) ratio of 2:1. This partially biobased epoxy system is composed of a liquid epoxy resin obtained as a reaction between bisphenol A diglycidyl ether (BADGE), alkyl glycidyl ether, and soybean oil resin with a viscosity of 600–700 cps. Regarding the hardener, it is a reaction adduct of ethylene amine, bisphenol A, benzyl alcohol, and soybean oil and has a viscosity of 60 cps.

To increase the contact area between the honeycomb core and the outer sheets, a polylactide (PLA) nonwoven mat was used. This was a spun-bonded PLA nonwoven for disposable clothes, supplied by RCfil NON-TEX (Vigo, Spain).

### 2.2. Manufacturing of PLA-Core Sandwich Structures

The development of the PLA-core sandwich structure was carried out in different stages: (a) Manufacturing of the hexagonal mold, (b) manufacture of the PLA-based honeycomb core, (c) manufacture of the PLA-flax outer sheets reinforced, and finally (d) bonding of the honeycomb core to the skins by means of adhesive joints as it is summarized in Figure 1.

The design and dimensions of the mold are of great relevance, as this will give the size of the hexagonal cells on honeycomb cores. High precision is required to ensure reproducibility in obtaining the cores, so that uniform honeycomb cores can be obtained. Figure 1a shows the detailed dimensions to obtain hexagonal cells. The aluminum blocks were machined with a WEISS WUM 100 universal rotary head-milling machine from DeTech (Łowicz, Poland) with numerical control. Two different stages were scheduled for the machining process: First, a roughing process with a 5 mm cylindrical milling tool with a spindle speed of 1660 rpm, a longitudinal feed rate of 55 mm min^−1^, and a plunging depth of 1 mm. After that, the second stage was carried out using a 60° conical milling tool with a spindle speed of 675 rpm, a longitudinal feed rate of 170 mm·min^−1^, and a plunging depth of 1 mm.

The next stage (Figure 1b) is the production of the PLA sheets; before further processing, PLA pellets were dried at 60 °C for 24 h due to its sensitivity to hydrolysis. PLA pellets were melted in a chill-roll unit equipped with a single-screw. The unit was a XTR series from Eurotech extrusion machinery SRL (Tradate, Italy). The selected temperature was 200 °C and the screw speed ranged between 43 and 55 rpm, depending on the final thickness: 43, 46, 51, and 55 rpm for 50, 150, 250, and 500 μm, respectively. The calendering rate was set to 0.9 m·min^−1^ for 50 and 100 µm thick sheets, while lower rates were used for thicker sheets, i.e., 0.2 m·min^−1^ for 250 μm and 0.1 m·min^−1^ for 250 μm. PLA sheets with a thickness of 50 and 100 μm show very high (50 μm) and high (100 μm) flexibility but (typical film thickness), in contrast, they do not offer handling stability. On the other hand, semi-hexagonal PLA sheets with 500 μm thickness are extremely brittle with very low flexibility. Finally, the semi-hexagonal PLA sheets with a thickness of 250 μm offered the best-balanced properties regarding geometry, flexibility, low brittleness, and good handling.

Additionally, Semi-hexagonal PLA films with thicknesses of 100, 250, and 500 m were subjected to a qualitative evaluation of the shape memory behavior, as shown in Appendix A (available in the Appendix A section).

Three different-thickness PLA honeycomb cores of 10, 20, and 30 mm were manufactured. The code used for the different structures was ‘PLA-HYNxx,’ where xx represents the height of each core. The semi-hexagonal sheets were obtained by hot compression molding using a Hoytom M.N 1417 hot compression machine supplied by Robima S.A. (Bilbao, Spain) at a temperature of 135 °C for 2 min with 8 tons of applied load. Finally, different semi-hexagonal sheets were bonded using Loctite^®^ 401 glue, to obtain the honeycomb core, as shown in Figure 1b.

The outer layers (faces or skins) consist of commingled polylactide (PLA) and natural flax fiber fabrics. Each outer skin is composed of two PLA/Flax commingled fabrics. These flexible fabrics were converted into a rigid laminate by hot compression molding at a temperature of 200 °C and an applied loaf force of 9 tons for 2 min, to allow PLA to melt and fully embed the aligned flax fibers, as shown in Figure 1c with an image of the flax/PLA commingled fabric before and after hot compression molding. 

Finally, the core and the skins were joined by means of an adhesive. For this purpose, a thermosetting adhesive layer was prepared. This adhesive layer consisted of a PLA-based nonwoven mat that was placed between the outer skins and the core, to increase the contact area between them and increase the retention of the EcoPoxy adhesive. The liquid fast-curing epoxy system was spread on the outer skin, and then a nonwoven layer was placed and embedded with a thin layer of the epoxy system using a hand lay-up technique. The same procedure was repeated to bond the outer skin. The final sandwich structure was placed into a press and subjected to slight force to ensure good contact between the honeycomb and the outer faces, while the epoxy adhesive resin crosslinked, leading to final composite sandwich structures with PLA honeycomb cores, as can be seen in Figure 1d.

### 2.3. Mechanical Characterization of PLA-Based Sandwich Structures

The mechanical properties of sandwich structures with PLA honeycombs were evaluated in flexural and compression conditions. Both were performed on a universal testing machine model ELIB 30 from Ibertest (Madrid, Spain). Flexural tests were carried out using a three-point bending configuration according to ASTM C393-00. The standard size of the samples for flexural tests was 150 mm in length, 50 mm in width, and a variable thickness (10, 20, and 30 mm), depending on the honeycomb thickness. The machine was equipped with a 5 kN load cell and the crosshead rate was set to 10 mm·min^−1^. The corresponding force–displacement graphs were collected, and from the results, it was possible to obtain the maximum load; besides, the core shear stress and the facing bending stress were calculated using Equations (1) and (2), respectively.
(1)$$\tau =\frac{P}{\left(d+c\right)\cdot b}$$
(2)$$\sigma_{b}=\frac{P\;\cdot \;L}{2\;\cdot \;t\;\cdot \;\left(d+c\right)\;\cdot \;b}$$
where *τ* is the core shear stress expressed in MPa, *P* is the maximum load expressed in N, *d* and *b* are the thickness and width of the sandwich structure, respectively, and *c* is the honeycomb thickness, all in mm. Regarding Equation (2), $\sigma_{b}$ is the facing bending stress expressed in MPa, *t* is the laminate thickness, *d* is the total sandwich thickness, *c* is the thickness of the honeycomb core, and *L* is the span length (constant value of 100 mm), all in mm.

Flatwise compressive tests were done according to ASTM C365/365M-16. This test was carried out on squared samples 50 mm length and width, and a variable thickness (10, 20, and 30 mm). A load cell of 50 kN with a crosshead rate of 10 mm min^−1^ was used. To obtain reliable data, at least 5 different specimens were tested for each sandwich structure.

## 3. Results

### Mechanical Properties of PLA-Based Sandwich Structures

The mechanical behavior of the PLA-based sandwich structures with a PLA honeycomb and PLA/flax outer layers was obtained in flexural and compression conditions. Obviously, as the thickness increases, the density of the panel (including the core and the skins) decreases (Table 1), mainly due to the PLA-honeycomb core. Some interesting properties of the shape memory behavior of PLA semi-hexagonal shapes are provided as Appendix A.

Figure 2 shows the characteristic force–displacement plots for flexural texts (three-point bending). These show the typical behavior of composite sandwiches in these conditions. An initial elastic stage (almost linear) up to a maximum load value (813.3 N for PLA-HYN10) can be seen. Then, the load decreases to values of 520–530 N. This drop is associated with debonding between the skin and the core as reported by Xu et al. [47]. A similar pattern can be observed for PLA-HYN20, with a maximum load after the elastic behavior of 1372.3 N, which is reduced to half after the first debonding signs (see Figure 2).

Figure 3 left column pictures show the elastic region for all three composite sandwiches as the corresponding force–displacement curves show. Clear signs of debonding (white ellipse) can be observed for PLA-HYN10 (right column). After this initial debonding, the load is shared by the core and the nondebonded skins until the core or the skin fails. As it can be seen in Figure 3, PLA-HYN-10 failure occurs by breakage of the core and the bottom face, while in PLA-HYN-20, failure occurs by progressive debonding (white ellipse) as its corresponding force–displacement curve suggests. Regarding the composite with a 30 mm PLA honeycomb, failure occurs in a different way. It seems that the core–skin adhesion is not as good as in other composites, and the core fails by shear at a relatively low force of 912.5 N. The failure image (Figure 3) shows a clear buckling-shear combined effect on the PLA honeycomb core that promotes early failure (white square). Similar results have been reported by Li et al. [48] in composite sandwiches subjected to out-of-plane loads, while Khan et al. [49] carried out an in-depth experimental study on the damage of honeycomb sandwich panels and compared the experimental results with those obtained by finite element analysis (FEA).

Table 2 summarizes the main results from both characterizations. It can be observed that the PLA-HYN10 sandwich structure, with a PLA honeycomb core thickness of 10 mm, shows the facing bending stress and the core shear stress with relatively high values of 33.0 and 0.66 MPa, respectively. These values are directly related to the maximum load the sandwich structure can withstand. These good results may be because the structure offers a good stability (no buckling in hexagonal cells) due to its low thickness compared to the 20 and 30 mm thickness honeycombs. In addition, the results suggest that the bonding between the components is good. Good bonding allows stress/load transfer by the shear between the outer faces and the core. The obtained results are comparable to those observed by Farooq et al. [17], with sandwich panels made from carbon fibers with an epoxy matrix (outer faces), Nomex^®^ honeycomb core and an epoxy-based adhesive film. Therefore, the mechanical properties of the PLA-based sandwich structures with PLA-based honeycombs could be considered a potential replacement for medium-to-high-performance structures based on synthetic materials. Cabrera et al. [50] produced polypropylene sandwich panels based on polypropylene composite laminates combined with a honeycomb polypropylene core, which presented a core shear stress and a face bending stress of 65% and 38%, respectively, lower than those obtained in this work for the same core thickness (20 mm).

The PLA-HYN20, with a 20 mm thick PLA-honeycomb structure, shows similar values of facing bending stress and core shear stress to PLA-HYN10. This means good load transfer from the faces to the core by shear. It should be noted that an increase in the core thickness results in a considerable increase in maximum load the sandwich structure can support, which is approximately 68% higher compared to the 10 mm thick core structure. Arbaoui et al. [51] and Xie et al. [52] suggested that by increasing the core thickness, the resulting sandwich structures allow better load transfer before they fail. This may be related to the fact that by increasing the core thickness, the final structure increases its stiffness-to-weight ratio. This is because the core structure mainly increases the bending moment of the structure by distancing the outer faces from the neutral axis and resisting shear loads. This also suggests excellent properties of the face-to-core bonding with the partially biobased epoxy resin.

Unlike the other structures, the PLA-HYN30 structure presents the lowest values concerning the core shear stress and facing bending stress, being approximately 53% lower in both cases, when compared to structures with a lower thickness (10 and 20 mm thick honeycombs). This can be caused by some separation that the outer faces had with the core, due to manufacturing issues decreasing the interface interaction between the core and the outer faces. This results in the stress not being correctly distributed and, subsequently, a decrease in the load that the structure can withstand, thus leading to a premature failure by a combination of shear and buckling [53]. In addition, Table 2 also includes the sample weight (W) and the ultimate load-to-weight (P/W) ratio, which agrees with the previous discussed results. These P/W ratios are very interesting and comparable to others reported in the literature with epoxy-carbon fiber (T700) skins (1 mm thick) and different cores (triangular and hexagonal panels, 15 mm thick). Xu et al. [47] reported higher core shear stress values up to 3 MPa as the skin contained carbon fiber and the cores, and was also fiber-reinforced. Nevertheless, the P/W ratios were lower than those obtained in this work, thus showing the interesting properties PLA-honeycombs can provide to composite structures in terms of mechanical properties and lightness.

Concerning the flatwise-compressive properties of the sandwich structures, Figure 4 shows the characteristic force–displacement curves, while the damaged samples after compression are gathered in Figure 5. The force–displacement curves show an elastic behavior before a maximum compression load is reached. This is close to 5620 N for the PLA-HYN10. Above this, the core collapses and, finally, when both skins are almost in contact, the force increases, as the densification stage is reached, as reported by Feng et al. [54]. Similar behavior can be observed for the other tested composites; nevertheless, the densification stage is delayed as the core thickness increases.

The main parameters from this test are summarized in Table 3; one can see that the PLA-HYN10 panel has a relatively low compressive strength value of 2.3 MPa. As with the flexural properties, it can be seen that when the core thickness is increased to 20 mm, the compressive strength has a slight increase value, reaching 2.6 MPa. Sallih et al. [55] suggested that in honeycomb structures, the core thickness does not have a great influence on its compressive properties, and is directly related to the thickness of the cell. This is because the compressive strength of the core is limited by the compressive strength of the original material. On the other hand, the structure with a 30 mm-thick core presents an unexpected drop in its compression properties. The compressive strength decreases by approximately 35% compared to the structure with a honeycomb PLA core 20 mm thick. This may be due to the lack of synergy between the core and the outer faces, causing the structure to lose stability, resulting in premature material failure, as detected by flexural characterization.

## 4. Conclusions

Through this study, it was possible to develop highly environmentally friendly sandwich structures based on the PLA honeycomb core and PLA/flax outer faces. The manufacturing process used in the core and in the outer layers provides good control over the size and shape of the hexagonal cells and the thickness of the layers, which is a first step in the reproducibility of this process. The use of a nonwoven PLA and epoxy resin as a bonding medium gave good adhesion between the core and the outer faces (mainly on PLA-honeycomb cores with 10 and 20 mm thickness), which is a critical issue in sandwich structures. Good control of the pressure exerted on the structure must be necessary to ensure sufficient contact between the PLA honeycomb and the outer PLA/flax faces. If the applied pressure is too high, it can lead the core to failure as observed in composites with 30 mm thick PLA honeycombs.

It was observed that a PLA honeycomb core with a height of 20 mm (wall thickness of 250 μm) offered excellent results in terms of flexural and flatwise compressive properties because the forces to which the structure is subjected can be appropriately transferred.

This work has demonstrated that PLA-based honeycombs can be interesting candidates to obtain environmentally friendly sandwich structures with good balanced mechanical properties for medium-to-high technological applications. These environmentally friendly composite sandwich panels can offer more than 95 wt.% biobased content, with balanced mechanical properties for engineering applications, which would greatly help to reduce the carbon footprint.

## Figures and Tables

**Figure 1 polymers-13-00342-f001:**
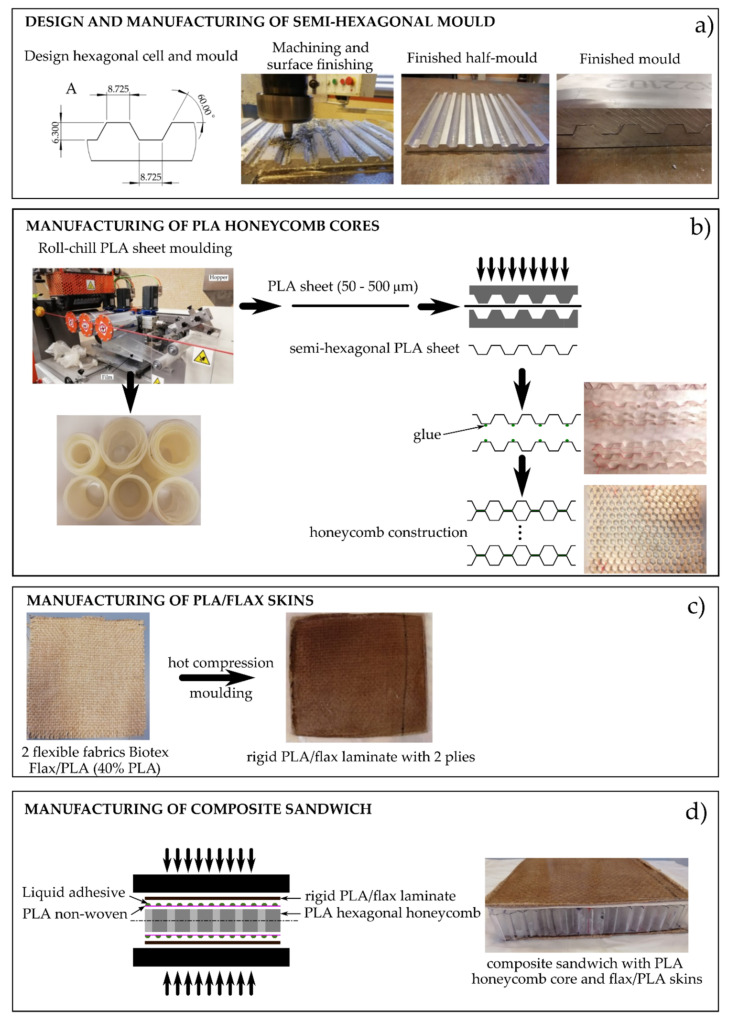
Schematic plot showing the different stages of the manufacturing process of (polylactide) PLA-honeycomb core-based sandwich structure.

**Figure 2 polymers-13-00342-f002:**
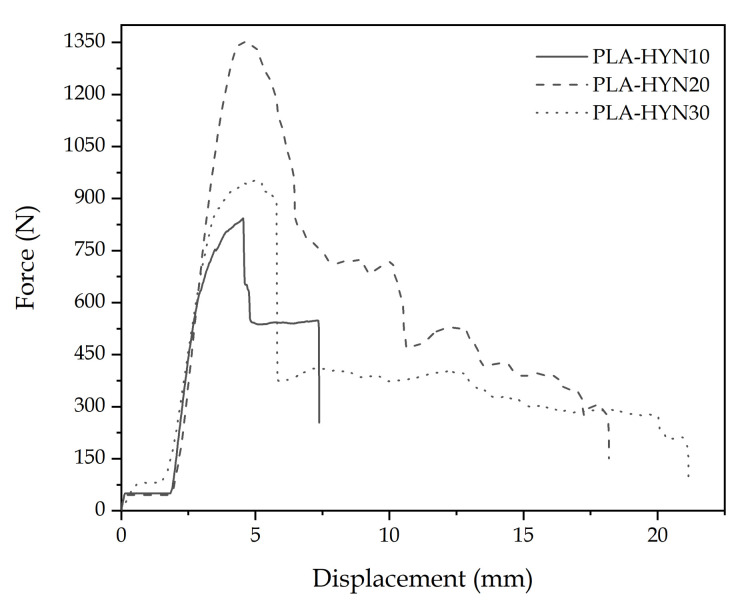
Characteristic force–displacement diagrams (three-point bending) for composite sandwiches with PLA honeycomb cores with different thickness.

**Figure 3 polymers-13-00342-f003:**
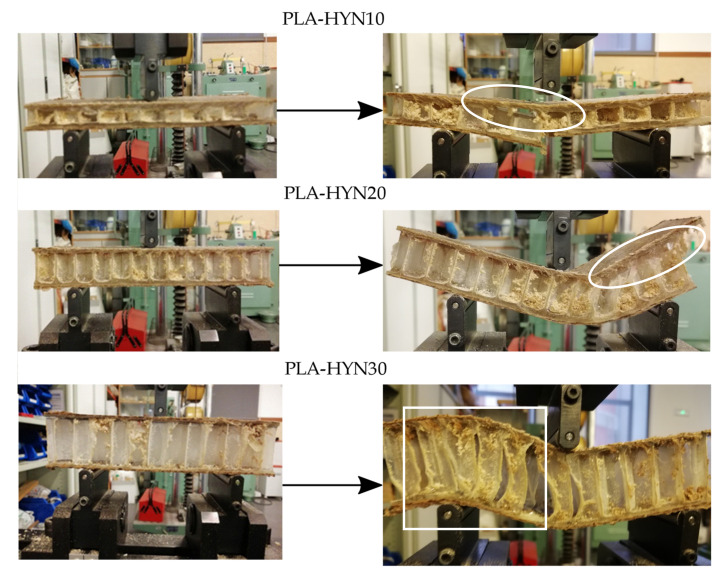
Pictures of the three-point bending test of composite sandwiches with PLA honeycomb cores with different thicknesses: The left column shows the initial elastic stage; the right column shows the image after the failure.

**Figure 4 polymers-13-00342-f004:**
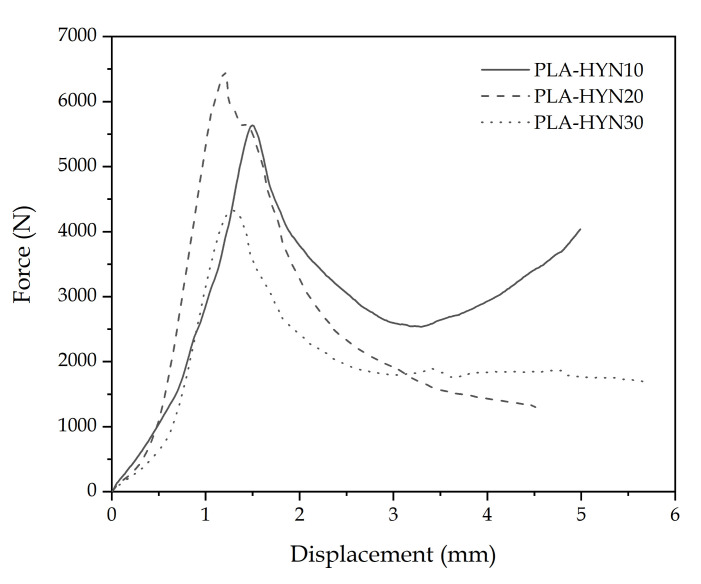
Characteristic force–displacement diagrams (flatwise compression test) for composite sandwiches with PLA honeycomb cores with different thicknesses.

**Figure 5 polymers-13-00342-f005:**
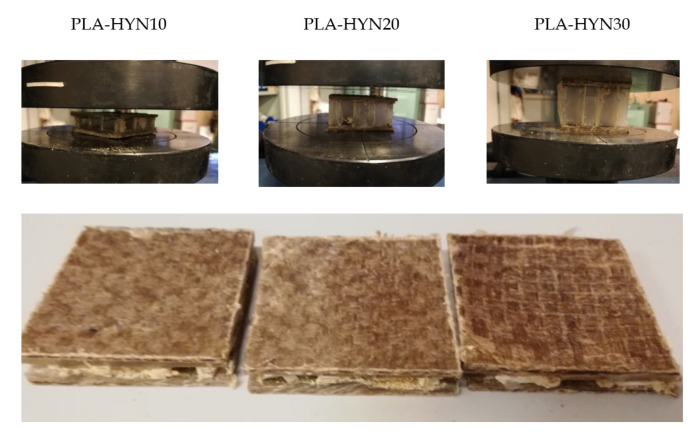
Pictures of the flatwise compression test of composite sandwiches with PLA honeycomb cores with different thicknesses: The left column shows the initial elastic stage; the right column shows the image after the failure.

**Table 1 polymers-13-00342-t001:** Equivalent density of the composite sandwiches with PLA honeycomb cores with different thicknesses.

Code	PLA Honeycomb Thickness (mm)	Equivalent Density of Composite Panel (Including Core and Skins) (kg·m^−2^)
PLA-HYN10	10	362
PLA-HYN20	20	279
PLA-HYN30	30	164

**Table 2 polymers-13-00342-t002:** Flexural (three-point bending) properties of PLA-based sandwich structures with PLA honeycomb cores.

Code	Flexural Properties				
	Ultimate Load, P (N)	Sample Weight, W (g)	P/W (N·g^−1^)	Core Shear Stress, τ (MPa)	Facing Bending Stress, $\sigma_{b}$ (MPa)
PLA-HYN10	813.3 ± 31.1	7.3 ± 0.3	111.4	0.66 ± 0.03	33.0 ± 1.7
PLA-HYN20	1372.3 ± 59.8	9.7 ± 0.8	141.5	0.63 ± 0.03	31.4 ± 1.4
PLA-HYN30	912.5 ± 55.6	11.7 ± 0.9	78.0	0.29 ± 0.02	14.7 ± 0.9

**Table 3 polymers-13-00342-t003:** Flatwise compressive properties of PLA-based sandwich structures with PLA honeycomb cores.

Code	Maximum Load, P (N)	Sample Weight, W (g)	P/W (N·g^−1^)	Compressive Strength, $\sigma_{c}$(MPa)
PLA-HYN10	5620 ± 300	18.0 ± 0.3	312.2	2.3 ± 0.3
PLA-HYN20	6419 ± 254	18.6 ± 0.5	345.1	2.6 ± 0.1
PLA-HYN30	4300 ± 420	21.1 ± 0.8	203.8	1.7 ± 0.1

## Data Availability

Not applicable.

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
