# Peer review of "Manufacturing and Characterization of Highly Environmentally Friendly Sandwich Composites from Polylactide Cores and Flax-Polylactide Faces"

_polymers, 2021, doi:10.3390/polym13030342_

Round 1
Reviewer 1 Report
The paper is interesting but there are too many English language mistakes, the authors must improve English quality before the paper may be accepted. Just a few examples: row 38 it is "top and bottom" not "upper and bottom", row 44 "load transfer" not "lead transfer", row 62 "because due" is wrong ... and so on, there are many many more.
On the more technical side, the paper is good, only I would not sound so triumphant in the Conclusions regarding the quality of the bonding adhesion. Indeed, as the authors coorectly noticed, PLA-HYN30 failed most probably because of separation between face sheets and core, thus I understand there is a bonding problem. Indeed, PLA-HYN30 had a maximum load that is slightly greater than PLA-HYN10, with a core that is three times as thick ... The authors should comment more on this.
Author Response
The paper is interesting but there are too many English language mistakes, the authors must improve English quality before the paper may be accepted. Just a few examples: row 38 it is "top and bottom" not "upper and bottom", row 44 "load transfer" not "lead transfer", row 62 "because due" is wrong ... and so on, there are many many more.
ANSWER
We thank the reviewer for this comment. As indicated, we have carried out an in-depth revision of the English Grammar and spelling, and all detected mistakes, have been corrected (yellow-marked) in the revised version.
On the more technical side, the paper is good, only I would not sound so triumphant in the Conclusions regarding the quality of the bonding adhesion. Indeed, as the authors correctly noticed, PLA-HYN30 failed most probably because of separation between face sheets and core, thus I understand there is a bonding problem. Indeed, PLA-HYN30 had a maximum load that is slightly greater than PLA-HYN10, with a core that is three times as thick ... The authors should comment more on this.
ANSWER
As the reviewer indicates we have reduced the triumphant sentences in the conclusions section since it is true that there is still a long way to solve this critical issue in composite sandwich structures. With regard to the maximum load of the PLA-HYN30 maximum load, compared to that of the PLA-HYN10, some additional comments and secondary literature have been provided in the revised version, but one of the simplest and more probable (and indicated in the work) reason is poor bonding.

Reviewer 2 Report
Polymers (ISSN 2073-4360)
Manufacturing and characterization of high environmentally friendly sandwich composites from polylactide cores and flax-polylactide faces
An excessive number of references. e.g., in lines 90 to 99 there are 14 references for a single paragraph which show consolidated knowledge information about PLA.
A single reference can be enough [27-29]. [30-33]..
An excessive number of the self-citations was used by the authors, as show below (13 self-citations)
- Lascano, D.; Valcárcel, J.; Balart, R.; Quiles-Carrillo, L.; Boronat, T. Manufacturing of composite materials with high environmental efficiency using epoxy resin of renewable origin and permeable light cores for vacuum-assisted infusion molding. Ingenius 2020, 62-73.
- Fombuena, V.; Sanchez-Nacher, L.; Samper, M.D.; Juarez, D.; Balart, R. Study of the Properties of Thermoset Materials Derived from Epoxidized Soybean Oil and Protein Fillers. J. Am. Oil Chem. Soc. 2013, 90, 449-457, doi:10.1007/s11746-012-2171-2.
- Carbonell-Verdu, A.; Bernardi, L.; Garcia-Garcia, D.; Sanchez-Nacher, L.; Balart, R. Development of environmentally friendly composite matrices from epoxidized cottonseed oil. European Polymer Journal 2015, 63, 1-10, doi:10.1016/j.eurpolymj.2014.11.043
- Lascano, D.; Quiles-Carrillo, L.; Balart, R.; Boronat, T.; Montanes, N. Toughened poly (lactic acid)— PLA formulations by binary blends with poly (butylene succinate-co-adipate)—PBSA and their shape memory behaviour. Materials 2019, 12, 622.
- Montava-Jordà, S.; Quiles-Carrillo, L.; Richart, N.; Torres-Giner, S.; Montanes, N. Enhanced interfacial adhesion of polylactide/poly (ε-caprolactone)/walnut shell flour composites by reactive extrusion with maleinized linseed oil. Polymers 2019, 11, 758.
- Lascano, D.; Moraga, G.; Ivorra-Martinez, J.; Rojas-Lema, S.; Torres-Giner, S.; Balart, R.; Boronat, T.; Quiles-Carrillo, L. Development of Injection-Molded Polylactide Pieces with High Toughness by the Addition of Lactic Acid Oligomer and Characterization of Their Shape Memory Behavior. Polymers 2019, 11, 2099
- Quiles-Carrillo, L.; Montanes, N.; Sammon, C.; Balart, R.; Torres-Giner, S. Compatibilization of highly sustainable polylactide/almond shell flour composites by reactive extrusion with maleinized linseed oil. Industrial Crops and Products 2018, 111, 878-888.
- Quiles-Carrillo, L.; Montanes, N.; Garcia-Garcia, D.; Carbonell-Verdu, A.; Balart, R.; Torres-Giner, S. Effect of different compatibilizers on injection-molded green composite pieces based on polylactide filled with almond shell flour. Composites Part B: Engineering 2018, 147, 76-85.
- Balart, J.; García‐Sanoguera, D.; Balart, R.; Boronat, T.; Sánchez‐Nacher, L. Manufacturing and properties of biobased thermoplastic composites from poly (lactid acid) and hazelnut shell wastes. Polymer Composites 2018, 39, 848-857.
- Quiles‐Carrillo, L.; Montanes, N.; Lagaron, J.M.; Balart, R.; Torres‐Giner, S. On the use of acrylated epoxidized soybean oil as a reactive compatibilizer in injection‐molded compostable pieces consisting of polylactide filled with orange peel flour. Polymer International 2018, 67, 1341-1351.
- Torres-Giner, S.; Montanes, N.; Fenollar, O.; Garcia-Sanoguera, D.; Balart, R. Development and optimization of renewable vinyl plastisol/wood flour composites exposed to ultraviolet radiation. Mater. Des. 2016, 108, 648-658, doi:10.1016/j.matdes.2016.07.037.
- Espana, J.M.; Samper, M.D.; Fages, E.; Sanchez-Nacher, L.; Balart, R. Investigation of the effect of different silane coupling agents on mechanical performance of basalt fiber composite laminates with biobased epoxy matrices. Polym. Compos. 2013, 34, 376-381, doi:10.1002/pc.22421
- Aguero, A.; Quiles‐Carrillo, L.; Jorda‐Vilaplana, A.; Fenollar, O.; Montanes, N. Effect of different compatibilizers on environmentally friendly composites from poly (lactic acid) and diatomaceous earth. Polymer International 2019, 68, 893-903.
Self-citation can help paper reviewers to check on the history of the author(s) and research topics. However, an excessive number of the self-citations demonstrate a particular way that some authors take to increase their personal impact factor.
The shape memory study of the material is simple and shallow. Thus, it was not justified. Upon point of view of the mechanical characterization it should be out.
- Abstract: specify in percentage the difference in the evaluated mechanical properties
- Lines 60-65: “In general, sandwich honeycomb structures are used because they provide
improved bending or flexural stiffness. Originally, sandwich structures with a honeycomb core
were only used in aeronautics and aerospace industries because due to their extremely
positive stiffness to weight ratio. Nevertheless, their production was laborious and, therefore,
expensive, which made them unsuitable for mass production. Aluminum has been, with
difference, the most common material for sandwich skins or cores in high performance
composites”.
This extract is poorly formulated and very generic. Please rewrite.
- Lines 72-73: “Today we can find sandwich structures made of monomaterials such as those
composed of PP sheets and PP honeycombs, achieving good uniformity”.
Please rewrite.
- Line 95: “thermos-compression”
maybe the authors want to say “Hot compression moulding”?
- Line 96-98: “The use of PLA in the manufacture of composite materials generates a
considerable decrease in greenhouse gases.”
Why is that? It is not clear.
- Line 105: “technical/engineering applications”. Why is that? It is not clear.
- Line 108: “Young´s modulus of up to 70 GPa and a tensile strength of 1.5 GPa”.
These are remarkably high values for natural fibers. Check in the literature used.
- “hot compression moulding” instead of “hot-press moulding”.
Materials and Methods
Make a schematic picture to show the manufacture of sandwich composites
item 3.1 must be part of item Materials and Methods and summarized (extensive and redundant).
Line 249: “a thermosetting adhesive layer was made ”in situ”. Explain.
- Line 258: “manufacturing” instead of “mmanufacturing”
- Lines 268-269: “because the structure offers a good stability due to its low thickness
compared to the 20 and 30 mm thickness honeycombs.” Explain.
Remark:
The mass of PLA cores also influences, it has not commented anything about the influence of this attribute.
An analysis of the plots of the flexural strength and compression tests are necessary.
Images of fractures of the specimens must be shown and evaluated.
The study of the shape memory of the material is very simple and was not justified. What is the scientific contribution of this result? This is not clear. Upon point of view of the mechanical characterization, it should be out.
Author Response
An excessive number of references. e.g., in lines 90 to 99 there are 14 references for a single paragraph which show consolidated knowledge information about PLA.
ANSWER
As the reviewer says, there are too many references in a single paragraph. We have gone into this and corrected it to avoid too many references in a single paragraph with general information about PLA.
A single reference can be enough [27-29]. [30-33]..
ANSWER
We are in total agreement with the reviewer regarding this comment. It is enough to show one reference. This has been corrected in the revised version. Regarding [17,27-29] references, only two have remained in the revised version, one about the use of PLA in the packaging industry and a second one related to its uses in biomedicine.
The same has been done with the pack of references [30-33] since all of them show different results of PLA-based composites with natural fillers. Only one reference has been kept in the revised version since it is enough to give an idea of this use of PLA.
An excessive number of the self-citations was used by the authors, as shown below (13 self-citations)
- Lascano, D.; Valcárcel, J.; Balart, R.; Quiles-Carrillo, L.; Boronat, T. Manufacturing of composite materials with high environmental efficiency using epoxy resin of renewable origin and permeable light cores for vacuum-assisted infusion molding. Ingenius 2020, 62-73.
- Fombuena, V.; Sanchez-Nacher, L.; Samper, M.D.; Juarez, D.; Balart, R. Study of the Properties of Thermoset Materials Derived from Epoxidized Soybean Oil and Protein Fillers. J. Am. Oil Chem. Soc. 2013, 90, 449-457, doi:10.1007/s11746-012-2171-2.
- Carbonell-Verdu, A.; Bernardi, L.; Garcia-Garcia, D.; Sanchez-Nacher, L.; Balart, R. Development of environmentally friendly composite matrices from epoxidized cottonseed oil. European Polymer Journal 2015, 63, 1-10, doi:10.1016/j.eurpolymj.2014.11.043
- Lascano, D.; Quiles-Carrillo, L.; Balart, R.; Boronat, T.; Montanes, N. Toughened poly (lactic acid)— PLA formulations by binary blends with poly (butylene succinate-co-adipate)—PBSA and their shape memory behaviour. Materials 2019, 12, 622.
- Montava-Jordà, S.; Quiles-Carrillo, L.; Richart, N.; Torres-Giner, S.; Montanes, N. Enhanced interfacial adhesion of polylactide/poly (ε-caprolactone)/walnut shell flour composites by reactive extrusion with maleinized linseed oil. Polymers 2019, 11, 758.
- Lascano, D.; Moraga, G.; Ivorra-Martinez, J.; Rojas-Lema, S.; Torres-Giner, S.; Balart, R.; Boronat, T.; Quiles-Carrillo, L. Development of Injection-Molded Polylactide Pieces with High Toughness by the Addition of Lactic Acid Oligomer and Characterization of Their Shape Memory Behavior. Polymers 2019, 11, 2099
- Quiles-Carrillo, L.; Montanes, N.; Sammon, C.; Balart, R.; Torres-Giner, S. Compatibilization of highly sustainable polylactide/almond shell flour composites by reactive extrusion with maleinized linseed oil. Industrial Crops and Products 2018, 111, 878-888.
- Quiles-Carrillo, L.; Montanes, N.; Garcia-Garcia, D.; Carbonell-Verdu, A.; Balart, R.; Torres-Giner, S. Effect of different compatibilizers on injection-molded green composite pieces based on polylactide filled with almond shell flour. Composites Part B: Engineering 2018, 147, 76-85.
- Balart, J.; García‐Sanoguera, D.; Balart, R.; Boronat, T.; Sánchez‐Nacher, L. Manufacturing and properties of biobased thermoplastic composites from poly (lactid acid) and hazelnut shell wastes. Polymer Composites 2018, 39, 848-857.
- Quiles‐Carrillo, L.; Montanes, N.; Lagaron, J.M.; Balart, R.; Torres‐Giner, S. On the use of acrylated epoxidized soybean oil as a reactive compatibilizer in injection‐molded compostable pieces consisting of polylactide filled with orange peel flour. Polymer International 2018, 67, 1341-1351.
- Torres-Giner, S.; Montanes, N.; Fenollar, O.; Garcia-Sanoguera, D.; Balart, R. Development and optimization of renewable vinyl plastisol/wood flour composites exposed to ultraviolet radiation. Mater. Des. 2016, 108, 648-658, doi:10.1016/j.matdes.2016.07.037.
- Espana, J.M.; Samper, M.D.; Fages, E.; Sanchez-Nacher, L.; Balart, R. Investigation of the effect of different silane coupling agents on mechanical performance of basalt fiber composite laminates with biobased epoxy matrices. Polym. Compos. 2013, 34, 376-381, doi:10.1002/pc.22421
- Aguero, A.; Quiles‐Carrillo, L.; Jorda‐Vilaplana, A.; Fenollar, O.; Montanes, N. Effect of different compatibilizers on environmentally friendly composites from poly (lactic acid) and diatomaceous earth. Polymer International 2019, 68, 893-903.
Self-citation can help paper reviewers to check on the history of the author(s) and research topics. However, an excessive number of self-citations demonstrate a particular way that some authors take to increase their personal impact factor.
ANSWER
We agree with the reviewer about this comment. We included references of the group related to some of the topics covered in this research work, but, as the reviewer indicates, it seems there are too many references and it is reasonable to leave out some of them as they do not provide particular relevant information for the project. Therefore, the number of self-citations has been reduced to the minimum to give an overview of the research topics developed in the group, related to the investigation. From the above-mentioned list, reference numbers: 13, 14, 16, 18, 19, 29, and 22 have been removed.
The shape memory study of the material is simple and shallow. Thus, it was not justified. Upon the point of view of the mechanical characterization, it should be out.
ANSWER
We thank the reviewer for this helpful point of view. Accordingly, we have removed the “qualitative assessment of shape memory behavior” from the manuscript and provided it as Supplementary Material.
- Abstract: specify in percentage the difference in the evaluated mechanical properties
ANSWER
As suggested by the reviewer, we have included numerical values and percentages in the abstract section to show the main features of the developed materials.
- Lines 60-65: “In general, sandwich honeycomb structures are used because they provide improved bending or flexural stiffness. Originally, sandwich structures with a honeycomb core were only used in aeronautics and aerospace industries because due to their extremely positive stiffness to weight ratio. Nevertheless, their production was laborious and, therefore, expensive, which made them unsuitable for mass production. Aluminum has been, with a difference, the most common material for sandwich skins or cores in high-performance composites”.
This extract is poorly formulated and very generic. Please rewrite.
ANSWER
We thank the reviewer for this comment. We have rewritten the paragraph to be more concise. We have also added secondary literature to support some statements.
- Lines 72-73: “Today we can find sandwich structures made of monomaterials such as those
composed of PP sheets and PP honeycombs, achieving good uniformity”.
Please rewrite.
ANSWER
As recommended by the reviewer, this sentence has been rewritten to give a clearer idea of composite panels manufactured with just one material, such as polypropylene.
- Line 95: “thermos-compression”
maybe the authors want to say “Hot compression moulding”?
ANSWER
This was a typo. As reviewer indicates, it refers to the “hot compression moulding” process. It has been corrected in the revised version.
- Line 96-98: “The use of PLA in the manufacture of composite materials generates a
considerable decrease in greenhouse gases.”
Why is that? It is not clear.
ANSWER
We thank the reviewer for this comment since it is not correct. It is true that PLA can be bioderived from starch-rich materials and hence, PLA-based composites can positively contribute to the environment from this point of view, compared to traditional composites with petroleum-derived components. Nevertheless, PLA, per se, is not enough to guarantee a decrease in greenhouse gases. Therefore, following the recommendations of the reviewer, the sentence has been rewritten to clearly show the role of PLA in obtaining environmentally friendly composites or green composites.
- Line 105: “technical/engineering applications”. Why is that? It is not clear.
ANSWER
We agree with the reviewer since the term “technical/engineering applications” is so generic. Therefore, we have included a paragraph showing some of the current uses of flax and other natural fibers as reinforcements in green composites, such as construction and building, automotive sector, sports industry.
- Line 108: “Young´s modulus of up to 70 GPa and a tensile strength of 1.5 GPa”.
These are remarkably high values for natural fibers. Check in the literature used.
ANSWER
We thank the reviewer for this comment and allow us to clarify this issue. These mechanical properties are not typical for a flax fiber. Mechanical properties of flax fibers are highly dependent on several factors such as the variety, fiber diameter, fiber length, crop conditions, and so on. For this reason, the mechanical properties of flax fibers change in a wide range. For example, Amroune et al. reported a change in the tensile strength from 1415 MPa to 431 MPa by changing the gauge length in the tensile test, thus giving clear evidence of the heterogeneity of mechanical properties flax fibers can provide. To avoid confusion, this sentence has been rewritten as a new paragraph showing these heterogeneities in mechanical properties. Moreover, some drawbacks related to the use of natural fibers as reinforcements have been included as they represent real challenges for green composites in the future.
- “hot compression moulding” instead of “hot-press moulding”.
ANSWER
We have revised the manuscript and all “hot-press moulding” terms have been change to the correct form “hot compression moulding”.
Materials and Methods
Make a schematic picture to show the manufacture of sandwich composites
item 3.1 must be part of item Materials and Methods and summarized (extensive and redundant).
ANSWER
We thank the reviewer for these helpful comments on this topic. The section has moved to “Materials and Methods” section and has been shortened to avoid redundant information. A schematic picture has been provided, which summarizes all the stages of composite panel manufacturing.
Line 249: “a thermosetting adhesive layer was made ”in situ”. Explain.
ANSWER
The term “in situ” meant that an adhesive layer composed of a PLA mat and an adhesive was placed between the skins and the core. But as the reviewer indicates, there is so much redundant information in this section. Therefore, this sentence has been removed since it does not give relevant information. In addition, a scheme summarizing all the stages has been provided, and all redundant information has been removed and rewritten.
- Line 258: “manufacturing” instead of “manufacturing”
ANSWER
This was a typo. It has been corrected.
- Lines 268-269: “because the structure offers good stability due to its low thickness
compared to the 20 and 30 mm thickness honeycombs.” Explain.
ANSWER
The explanation about this issue has been changed to avoid confusion.
Remark:
The mass of PLA cores also influences, it has not commented anything about the influence of this attribute.
ANSWER
We agree with the reviewer about this issue. To overcome this, we have included the density of the composite sandwiches, as well as the sample, weigh in flexural and flatwise compression test. With this information, we have provided the values of the maximum load (P) to the weight (W) ratio and commented on this.
An analysis of the plots of the flexural strength and compression tests are necessary.
Images of fractures of the specimens must be shown and evaluated.
ANSWER
We have provided and commented on the force-displacement curves for both tests (three-point bending and flatwise compression) in the revised version. The curves have been compared with some others reported in the literature.
The study of the shape memory of the material is very simple and was not justified. What is the scientific contribution of this result? This is not clear. Upon the point of view of the mechanical characterization, it should be out.
ANSWER
We thank the reviewer for this helpful point of view. It is true that the shape memory behaviour of PLA does not give any relevant information about the mechanical properties of composite sandwich structures. Accordingly, we have removed the “qualitative assessment of shape memory behavior” from the manuscript, and provided it as “Supplementary information”.

Reviewer 3 Report
- Reviewer’s summary and comments
The manuscript presents a manufacturing process and mechanical properties of a lightweight honeycomb sandwich structure which was made of polylactide (PLA) cores and flax-polylactide faces. The sandwich structure was considered to be a friendly composite that can be used in the fields of packaging and biomedicine. The authors presented the mechanical properties of the sandwich structure using a three-point bending test and a flatwise compressive test. Besides, they also observed the shape memory performance of honeycomb cores.
The manuscript focuses on presenting the manufacturing process of the sandwich structure which included the manufacturing of the core and the outer faces. I think the author may consider moving the section “3.1 Manufacturing of PLA-core sandwich structures” to the section “2 Material and methods”.
Some discussions on mechanical properties and shape memory performance should be performed.
The manuscript has a very thorough reference list.
It is an interesting work. So, the reviewer recommends a publication of this manuscript with minor revision.
- In the very first of the Introduction, the author may consider adding more references regarding applications of sandwich structures. For example, [1] doi: https://doi.org/10.3390/ma13173791 for packaging application and [2] doi: https://doi.org/10.3390/app9245541 for aerospace application.
- Pages 2-3, line 95-96, it is required to present recent papers that used PLA structures in the fields of packaging and biomedicine. This helps the readers to have the most updating information.
- Page4, line 164, the ASTM C365/365M-05 standard test may be superseded. Please use C365/365M-16. Also, please explain why the authors used the rate of 10 mm/min for both mechanical testing methods. It may be a bit faster than the recommended one.
- Page 4, Line 174, the authors wrote “the switching transition temperature was set at 135 °C (close to PLA cold crystallization temperature)”. The reviewer wonder why the author did not use the glass transition temperature (Tg: 50-60 °C)?
- Section 3.2, please include the force-displacement curve for each tested structure.
- To point out the good mechanical properties of the proposed structure, the author may consider calculating the specific core shear stress and the specific facing bending stress. Then, making comparisons with other structures in the literature (if they are available)
- It is required to show the failure maps of the tested structures.
- The shape memory performance of the core is presented with different thicknesses of the core. The author should address the role of shape recovery in a practical application such as packaging and biomedicine fields.
Author Response
The manuscript presents a manufacturing process and mechanical properties of a lightweight honeycomb sandwich structure which was made of polylactide (PLA) cores and flax-polylactide faces. The sandwich structure was considered to be a friendly composite that can be used in the fields of packaging and biomedicine. The authors presented the mechanical properties of the sandwich structure using a three-point bending test and a flatwise compressive test. Besides, they also observed the shape memory performance of honeycomb cores.
The manuscript focuses on presenting the manufacturing process of the sandwich structure which included the manufacturing of the core and the outer faces. I think the author may consider moving the section “3.1 Manufacturing of PLA-core sandwich structures” to the section “2 Material and methods”.
ANSWER
We thank the reviewer for this helpful comment on this topic. According to it, the section has moved to “Materials and Methods” section and has been summarized to avoid redundant information. A schematic picture has been provided, which summarizes all the stages of the composite panel manufacturing in a simple way.
Some discussions on mechanical properties and shape memory performance should be performed.
ANSWER
As indicated by the reviewer, an in-depth analysis and discussion of the obtained results has been provided in the revision version.
The manuscript has a very thorough reference list.
ANSWER
Some unnecessary references have been removed and other relevant ones have been added.
It is an interesting work. So, the reviewer recommends a publication of this manuscript with minor revision.
In the very first of the Introduction, the author may consider adding more references regarding applications of sandwich structures. For example, [1] doi: https://doi.org/10.3390/ma13173791 for packaging application and [2] doi: https://doi.org/10.3390/app9245541 for aerospace application.
ANSWER
We thank the reviewer for this helpful comment. It is interesting to show the readers some of the actual uses of composite sandwich structures in different sectors. According to this, a new paragraph has been added at the very first part of the “Introduction” section with special emphasis on current applications of sandwich structures. Both recommended references have been added, as well as some others related to the automotive industry, civil infrastructure, and so on.
- Pages 2-3, line 95-96, it is required to present recent papers that used PLA structures in the fields of packaging and biomedicine. This helps the readers to have the most updating information.
ANSWER
We agree with this comment. Accordingly, recent references about the use of PLA on both biomedicine (3d printed resorbable parts and scaffolds) and packaging industry (with antioxidant properties and for active packaging) have been provided in the revised version.
- Page4, line 164, the ASTM C365/365M-05 standard test may be superseded. Please use C365/365M-16. Also, please explain why the authors used the rate of 10 mm/min for both mechanical testing methods. It may be a bit faster than the recommended one.
ANSWER
Following the recommendations of the reviewer, we have changed the ASTM code to the correct one, i.e. C365/365M-16. Regarding the crosshead speed rate, as the reviewer has noted, 10 mm/min is a little bit faster than the recommended values. We will take this into account for future works with these sandwich-type structures. Anyway, since all composite panels are tested with the same conditions, the obtained values can be compared, but we will take this into account for future works, since results could be different with less aggressive conditions.
- Page 4, Line 174, the authors wrote “the switching transition temperature was set at 135 °C (close to PLA cold crystallization temperature)”. The reviewer wonder why the author did not use the glass transition temperature (Tg: 50-60 °C)?
ANSWER
The reviewer is right about this. In fact, this is a mistake while writing the manuscript. The trigger temperature for the shape memory behaviour was 60 ºC as reported in the included references. The use of higher temperatures of about 135 ºC would not be as good for the shape memory since cold crystallization would simultaneously occur. We thank the reviewer for advising this important mistake that has been corrected in the revised version (in fact, we used 70 ºC which is 10 ºC above the Tg, but remarkably lower than the cold crystallization). As the shape memory recovery of PLA is not the focus of the manuscript, this section has been provided as “Supplementary information” with the corresponding corrected data.
- Section 3.2, please include the force-displacement curve for each tested structure.
ANSWER
As indicated by the reviewer, the force/stress-displacement curves have been provided and commented in the revised version as they give additional information on how failure occurs.
- To point out the good mechanical properties of the proposed structure, the author may consider calculating the specific core shear stress and the specific facing bending stress. Then, making comparisons with other structures in the literature (if they are available).
ANSWER
In accordance with this comment, we have been searching for the way other authors represent the relationship between mechanical properties and weight. One of the most used parameters is the ultimate load (P) to sample weight (W) ratio. We have included the density of the composite sandwiches as well as the sample weigh in flexural and flatwise compression test. With this information we have provided the values of the maximum load (P) to the weight (W) ratio and commented about this.
- It is required to show the failure maps of the tested structures.
ANSWER
The failure images of the different samples have been provided in the revised version since, as the reviewer suggests, they can provide relevant information on how failure occurs.
- The shape memory performance of the core is presented with different thicknesses of the core. The author should address the role of shape recovery in a practical application such as packaging and biomedicine fields.
ANSWER
We are in total agreement with the reviewer about the sole of shape recovery in different sectors. Subsequently, some applications in the packaging and biomedicine have been provided and commented as supplementary information.

Round 2
Reviewer 2 Report
Title: Manufacturing and characterization of high environmentally friendly sandwich composites from polylactide cores and flax-polylactide faces (polymers-1052044) Dear Sirs, The recommended reviews were performed. The topic is interesting and paper is quite satisfactory. Thus, I recommend it for publication. Best Regards.